# Comparison of Serum Selenium, Homocysteine, Zinc, and Vitamin D Levels in Febrile Children with and without Febrile Seizures: A Prospective Single-Center Study

**DOI:** 10.3390/children10030528

**Published:** 2023-03-09

**Authors:** Emrah Çığrı, Funda Çatan İnan

**Affiliations:** Faculty of Medicine, Kastamonu University, Kastamonu 37150, Turkey; fcatan@kastamonu.edu.tr

**Keywords:** febrile seizure, selenium, zinc, vitamin D, vitamin B12

## Abstract

Objective: Febrile seizure is a complication that makes physicians and families uneasy when detected in children with a high fevers. This study aimed to compare children with febrile seizures and children without seizures in blood selenium, zinc, homocysteine, vitamin D, vitamin B12, and magnesium levels. Materials and Methods: The study group included sixty-one children between the ages of 1–5 who came to the pediatric emergency department with febrile seizure. The control group had 61 children with fever without seizure, who were compatible with the study group in age, sex, and elapsed time since the onset of fever. Blood samples were taken from the patients during their admission. Selenium, zinc, vitamin D, homocysteine, vitamin B12, and magnesium levels were measured, and the data of the two groups were compared. Additionally, patients in the study group had two subgroups, simple and complex febrile seizures, and their parameters were compared. Results: Selenium, zinc, vitamin D, and vitamin B12 levels were significantly lower in the study group than in the control group (*p* < 0.001), and there was no significant difference in homocysteine (*p* = 0.990) and magnesium levels (*p* = 0.787) between the two groups. Moreover, no significant difference was found between those with simple and complex febrile seizures in selenium, vitamin D, homocysteine, vitamin B12, and magnesium levels. Conclusions: Elevated levels of selenium, zinc, vitamin D, and vitamin B12 in the blood of children with fevers help to prevent febrile seizures.

## 1. Introduction

Febrile seizure (FS) is a seizure accompanied by fever (>38 °C or >100.4 °F) due to any focus infection other than central nervous system (CNS) in children aged six months to five years without a history of febrile seizures or CNS anomaly [1]. Febrile seizures are not related to the rate of rise in fever but related to length of the fever [2]

The febrile seizure is the most common type of seizure in children and occurs in about 2–5% of children [3]. Etiology is thought to be multifactorial; especially certain vaccines that cause high fever (e.g., MMR) and viral infections (e.g., HHV-6), history of FS in first or second-degree relatives, long-term stay in the neonatal intensive care unit, mother’s smoking, pyridoxine (vitamin B6) deficiency, and trace element disorders such as iron, selenium, and zinc are the main risk factors for FS [4,5]. Certain genes may also increase the risk for familial epilepsy syndromes that cause febrile seizures. They do this by increasing sensitivity to environmental risk factors [6,7].

Febrile seizures can be simple and complex FS. FS is simple if:it does not show focal characteristics,it lasts less than 15 min,it does not recur within 24 h.

On the other hand, the FS is complex if: it shows focal characteristics,it lasts more than 15 min,it recurs within the first 24 h [8].

Febrile seizures are generally benign, but according to recent studies: (1) they can cause psychiatric diseases and even sudden death, (2) they can cause brain damage if prolonged, and (3) patients have a 14% chance of developing epilepsy within the first 2 years after febrile status [9]. If the first febrile seizure occurred before 18 months, the risk of recurrence within 2 years is between 15–70%. The recurrence rate within one year after initial febrile status is 16% (95% CI 10–24), and the mortality rate after 8 years is 0% [10,11].

Studies were indecisive regarding whether trace elements and vitamin levels in the blood affect the development of FS. Therefore, this study aimed to investigate how trace elements, such as selenium and zinc in the blood, vitamin D, and homocysteine levels influence the development of FS in children with fever. 

## 2. Materials and Methods

To estimate the required sample size, we used G*Power [12] to conduct a statistical power analysis with α = 0.05, power = 0.95, and 0.6 high effect size. Informed consent was obtained from all subjects involved in the study, as shown in the back matter.

This research had a study group and a control group. The study group included sixty-one children who applied to the Pediatric Emergency Clinic of Kastamonu Training and Research Hospital between October 2022 and January 2023 with FS between the ages of 1–5. The control group had sixty-one children who had a fever but did not have seizures and were similar to the study group characteristics in age, sex, and elapsed time since the onset of fever formed the control group. Children in the study groups were in two subgroups: simple FS and complex FS. There were comparisons between the study group, control group, and two subgroups of the study group in serum selenium, zinc, homocysteine, and vitamin D levels. The criteria to differentiate simple and complex FS are in Table 1 [8,13].

Seizures resulting from high fever (>38 °C) due to an infection other than CNS were diagnosed with FS [1]. Those with CNS anomalies, those with congenital metabolic diseases, those with heart disease, those with epilepsy, and those with malnutrition were not in the study. 

All blood analyses were performed from the first blood samples taken after the informed consent form was obtained at the time of admission. Selenium level was measured in serum with Perkin Elmer Atomic Absorption method (PinAAcle 900 Z), zinc level with homogeneous colorimetric enzyme technique (SATURNO 300), homocysteine level with chemiluminescence immunoassay (CLIA) method in plasma with EDTA (Siemens Immulite 2000, Siemens Healthcare Diagnostics, Llanberis, UK), Vitamin B12 level with Competitive binding immunoenzymatic assay (Beckman Coulter Inc., Brea, CA, USA), 25-OH vitamin D level with CLIA method (Beckman Coulter Inc.; Brea, CA, USA), and complete blood count with Automatic hematological analyzer (XN-1000-Hematology-Analyzer-Sysmex Corporation, Kobe, Japan).

Statistical Analysis: The significance of differences between the means of the continuous variables was determined by the Mann–Whitney U test for non-normally distributed data [14]. Pearson’s chi-square test was employed to determine the association between two categorical variables [14]. None of the continuous variables showed a normal distribution, and they were described as their median values. Statistically significance was accepted at a level of *p* < 0.05. All data were analyzed using SPSS 26.00 (SPSS Inc., Chicago, IL, USA). 

Ethical Approval: The study was approved by the Non-Interventional Ethics Committee of Kastamonu University with the decision number 2022-KAEK-107, date 19 October 2022. 

## 3. Findings

Our study consisted of 122 children, a study group of sixty-one children with FS, and a control group of sixty-one children with fever but without seizures. There was no significant difference between the demographic characteristics of the children in the study and control groups (*p* > 0.05) (Table 2).

The children in the study group had the following health condition issues: 33 (54.1%) of the children had simple FS, 28 (45.9%) had complex FS, 68.8% had a family history of FS or epilepsy, 17 children had lower respiratory tract infection (LRTI), 12 children had upper respiratory tract infection (URTI), 10 children had acute tonsillitis, 10 children had acute otitis media (AOM), 8 children had urinary tract infection (UTI), and 4 children had acute gastroenteritis (AGE) (Figure 1). On the other hand, of the children in the control group, 14 children had URTIs, 14 children had UTIs, 13 children had acute tonsillitis, 11 children had AOM, 8 children had LRTIs, and 1 child had AGE (Figure 2).

Table 3 presents the serum level of biochemical characteristics of children in the study group and control group. Regarding the differences in the biochemical parameters of the children in the study and control groups, the median values of zinc, selenium, vitamin B12, and vitamin D levels were significantly lower in the study group than in the control group (*p* < 0.001). However, there was no significant difference between the two groups in homocysteine (*p* = 0.990) and magnesium levels (*p* = 0.787).

Table 4 summarizes biochemical findings with simple febril seizure and complex febril seizure. According to the subgroups of simple and complex FS, there were no significant differences in biochemical parameters between the two groups.

Table 5 shows that statistically significant differences were found between Red cell Distribution Width (RDW) (*p* < 0.001) and Immature Granulocyte (IG) (*p* = 0.01) with respect to two groups. The value of RDW was higher in patients with febril seizure while the value of IG was higher in patients without febril seizure. 

## 4. Discussion

We conducted a study to determine what changes were present in serum selenium, zinc, homocysteine, and vitamin D levels in children aged 1–5 years who experienced febrile seizures. The levels of selenium, zinc, and vitamin D were significantly lower in children with FS than in children without seizures, but there was no significant difference in homocysteine levels. Furthermore, there was no significant difference in serum selenium, zinc, vitamin D, and homocysteine levels in children with simple and complex FS.

Selenium is a trace element, and it is the main component of enzymes such as glutathione peroxidase, deiodinases, and thioredoxin reductase with the following critical roles in muscle function, reproduction, tumor suppression, and antioxidant effects, particularly in brain cells [15]. It is also necessary for the regular functioning of the immune system because selenium stimulates helper T cells along with natural killer and cytotoxic T cells and plays a significant role in antibody formation and phagocytosis [16]. 

Selenium deficiency had been effective in the pathogenesis of epilepsy in several studies, but its role in FS has not been determined [17,18]. In the study of Bakhtiari et al. [19], those with FS in children older than one year had significantly lower selenium levels than children without seizures. Similarly, Amiri et al. [20] and Mahyar et al. [21] found that the selenium levels of children with FS were significantly lower than those of febrile children without seizures. Additionally, the serum selenium levels of children with FS are statistically significantly lower than selenium levels a few months after recovery [22,23]. Jill et al. [24] stated that children with lower serum selenium levels were more prone to psychiatric disorders such as anxiety disorder, generalized anxiety disorder, school phobia, panic/somatic disorder, and social anxiety disorder, compared to other children. On the other hand, Khoshdel et al. [25] found that children with FS had lower selenium levels than children without seizures, but this was not significant. Again, Safaralizadeh et al. [26] found no significant difference in selenium levels between children with FS and children with fever without seizures. In our study, similar to most of the literature, serum selenium levels of children with FS were significantly lower than those of febrile children without seizures.

Zinc prevents excitatory neuronal discharge and seizure by the following two mechanisms: (1)increasing the effect of glutamic acid decarboxylase, a rate-limiting enzyme in the synthesis of an inhibitory neurotransmitter in CNS, GABA (gamma-aminobutyric acid),(2)facilitating the suppression of an excitatory neurotransmitter in CNS, NMDA (N-methyl D-aspartate) by calcium [27].

However, increased cytokines during FS reduce serum zinc levels by stimulating the production of metallothionein. Metallothionein accelerates the passage of zinc to the liver [28]. In many studies, serum zinc levels of children who come with FS are significantly lower than children with fever who do not have seizures [29,30,31]. Furthermore, Ganesh et al. [32] stated that children with epilepsy had lower serum zinc levels than children without epilepsy. On the other hand, Çelik et al. [33] and Cho et al. [34] reported no significant difference in zinc levels in children with FS compared to febrile children without seizures. In our study, zinc levels were lower in children with FS than in the control group, similar to most of the literature in this respect.

Vitamin D acts as both a neurotransmitter and a brain protector in CNS, and the positive effect of vitamin D on epilepsy patients is well-known [35,36]. Kalueff et al. [37] reported that the seizure threshold of mice with vitamin D delivered to the hippocampus region of their brain increased, while the seizure threshold of mice with vitamin D receptor deactivated in their brain decreased, and their seizure sensitivity increased. In their study, Al Khalifah et al. [38] defined the relationship between vitamin D deficiency and epilepsy and reported that vitamin D supplementation may reduce requiring multiple drugs in treating epilepsy.

The role of vitamin D in FC remains unclear, although there are limited studies in the literature. In their study, Bhat et al. [39] found that there was a significant relationship between vitamin D level and FS and that there was a negative correlation between FS relapse and vitamin D level (r = −0.672, *p* < 0.001), and they suggest that vitamin D could also be used in the treatment of simple recurrent FS. Similarly, Singh et al. [40] and Motlaghzadeh et al. [41] stated that vitamin D deficiency was present at a higher rate among children with FS. On the other hand, Heydarian et al. [42] reported that there was no statistically significant relationship between serum vitamin D level and FS formation. In our study, there was a significant relationship between low vitamin D and FS manifestation, consistent with most. 

Homocysteine is a non-essential amino acid synthesized from the amino acid methionine. Lifestyle, diet, medications, and genetic factors can influence homocysteine levels in humans. High homocysteine levels are a well-known risk factor for neurovascular diseases such as epilepsy, dementia, migraine, and cardiovascular diseases [43]. 

Homocysteine has a toxic effect on CNS through the following mechanisms: DNA damage, triggering apoptosis, and oxidative stress [43]. However, few studies investigated the relationship of homocysteine with epilepsy and FS, so the relationship between homocysteine level and FS remains unclear. In the study of Biancheri et al. [44] and Telfeian et al. [45], homocysteine triggered seizures in mice. On the other hand, Özkale et al. [46] found no significant difference in homocysteine levels between children in the control group and children with FS, and homocysteine level was not an independent risk factor for FS. In our study, no significant difference was present between children with FS and children in the control group in terms of homocysteine levels.

Vitamin B12 is a water-soluble vitamin that must be taken in the diet, and it is effective in growth and development in humans. The only dietary source of vitamin B12 is animal foods [47]. Vitamin B12 deficiency in children is caused by disturbances in the metabolism and transport of B12, its abnormal absorption, and its low dietary intake [48]. In children with vitamin B12 deficiency, methylmalonyl CoA cannot be converted to succinyl CoA and the accumulated methylmalonyl CoA is used in the synthesis of fatty acids. As a result, myelin synthesis is impaired in the central nervous system and children’s cognitive performance and brain development are adversely affected [49]. 

Although it is thought that low vitamin B12 may be a factor triggering seizures, its mechanism of action is still unclear [50,51]. Few studies determined the relationship between vitamin B12 deficiency and febrile seizures in children. Osifo et al. [50] found that the levels of vitamin B12 were significantly higher in children with febrile seizures compared to children with FS, but there were no significant differences in vitamin B12 levels in the cerebrospinal fluid. Biancheri et al. [44] determined EEG abnormalities in all children. Ozkale et al. [46] found that vitamin B12 levels were significantly lower in children with febrile seizures compared to the control group. In a research study, vitamin B12 levels were significantly lower in children with depression compared to children without depression [52]. In addition, Altun et al. [53] mentioned that vitamin B12 levels were significantly lower in children with attention deficit hyperactivity disorder compared to those without. In our study, similar to the majority of the literature, vitamin B12 levels were found to be significantly lower in children with febrile seizures compared to those without.

RDW is a parameter that measures the differences in volume and size of circulating erythrocytes. In other words, the larger the size of the erythrocytes, the greater their heterogeneity [54]. The usage of RDW in clinical practice has been limited by determining the etiology of anemia for a long time. However, recent studies have found that RDW has a decisive effect in predicting all-cause mortality in patients with severe diseases or hospitalized in the intensive care unit, and it is closely related to the severity and prognosis of infectious diseases [55,56]. Göksugur et al. [57] stated that RDW is a significant marker in the differentiation of febrile seizure types. Yıldız et al. [58] reported that RDW was not significant in the differentiation of febrile seizure types. Örnek et al. [59] compared the complete blood count parameters of children with febrile seizures and children with fever but not seizures. Moreover, they stated that hemoglobin, hematocrit, mean platelet volume, neutrophil lymphocyte ratio, and platelet lymphocyte ratio were significantly different, while other parameters were not significant. In our study, there were significant differences in RDW and IG values of complete blood count parameters between the two groups.

## 5. Conclusions

High serum selenium, zinc, vitamin D, and vitamin B12 levels prevent the development of seizures in children between the ages of 1–5 in the pediatric emergency department due to fever.

## Figures and Tables

**Figure 1 children-10-00528-f001:**
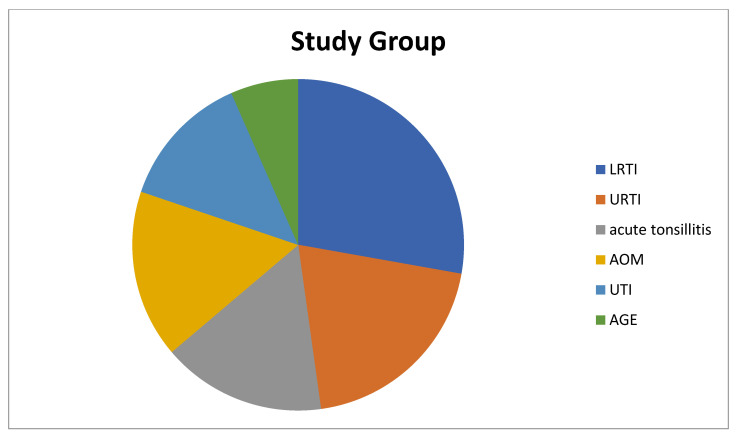
Diagnoses of patients in the study group that cause fever. LRTI: Lower Respiratory Tract Infection, URTI: Upper Respiratory Tract Infection, AOM: Acute Otitis Media, UTI: Urinary Tract Infection, AGE: Acute Gastroenteritis.

**Figure 2 children-10-00528-f002:**
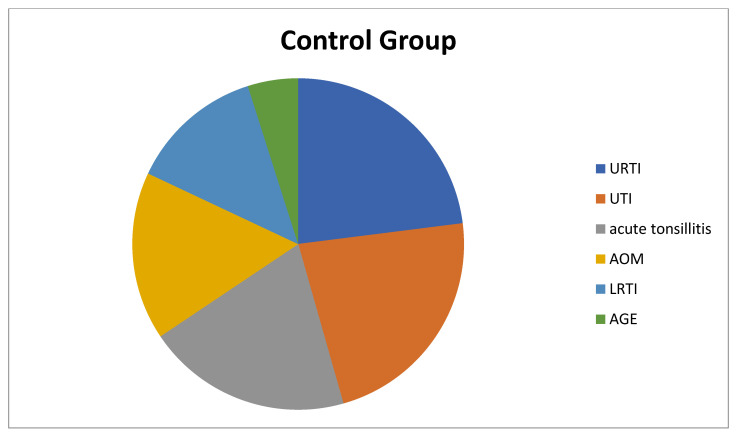
Diagnoses of patients in the control group that cause fever. LRTI: Lower Respiratory Tract Infection, URTI: Upper Respiratory Tract Infection, AOM: Acute Otitis Media, UTI: Urinary Tract Infection, AGE: Acute Gastroenteritis.

**Table 1 children-10-00528-t001:** Diagnostic criteria used in the differentiation of simple and complex febrile seizures.

Simple Febrile Seizure	Complex Febrile Seizure
Lasts less than 15 min	Lasts more than 15 min
Having a generalized seizure	Having a focal seizure
No neurological findings after the seizure	Neurological findings after the seizure (Todd paralysis)
No repetition of seizures within 24 h	Seizure recurrence in the first 24 h

**Table 2 children-10-00528-t002:** Demographic characteristics of children in the study and control groups.

	Study Group	Control Group	*p*-Value
Gender (Girls/Mens)	23/38	23/38	1.000
Age (months)	35.36 ± 13.91	35.54 ± 13.87	0.943
Fever (°C)	38.86 ± 0.52	38.84 ± 0.53	0.892
The time between the onset of fever and admission (hours)	33.44 ± 9.69	33.34 ± 9.68	0.955

**Table 3 children-10-00528-t003:** Comparison of biochemical parameters of children in the study and control groups.

Variables	Total (*n* = 122)	Study Group *n* = 61 (50%)	Control Group *n* = 61 (50%)	*p*-Value
Vitamin D (ng/mL)	15.4 (10.47–15.45)	12.8 (10.6–15.8)	19.6 (3.7–31.5)	**<0.001**
Selenium (µg/L)	71.0 (50.75–91.25)	51.0 (35.5–66.5)	91.0 (75.5–106.5)	**<0.001**
Homocysteine (mmol/L)	75.7 (64.27–87.37)	75.7 (64.2–87.5)	73.9 (61.2–85.3)	0.990
Zinc (µg/dL)	74.0 (68.87–84.50)	67.6 (62.15–72.35)	80.2 (73.70–94.05)	**<0.001**
Mg (mg/dL)	19.9 (13.27–21.7)	19.9 (96.5–217.0)	19.9 (9.65–21.7)	0.787
Vitamin B12 (pg/mL)	259.0 (216.25–315.0)	225.0 (174.0–246.5)	315.0 (276.0–341.5)	**<0.001**

Mann–Whitney U-test (median of variables) Mg: Magnesium.

**Table 4 children-10-00528-t004:** Comparison of biochemical and complete blood count parameters of children with simple FS and complex FS.

Variables	Simple Febrile *n* = 33 (54.1%)	Complex Febrile *n* = 28 (45.9%)	*p*-Value
Vitamin D (ng/mL)	13.8 (10.95–16.3)	11.75 (10.45–15.35)	0.193
Selenium (µg/L)	47.0 (32.5–63.0)	53.0 (43.25–69.5)	0.136
Homocysteine (mmol/L)	77.1 (60.45–86.85)	75.40 (64.32–88.27)	0.885
Zinc (µg/dL)	67.0 (62.2–71.95)	70.15 (66.15–74.27)	0.548
Mg (mg/dL)	18.9 (2.4–21.7)	20.25 (19.1–21.6)	0.195
Vitamin B_12_ (pg/mL)	225.0 (173.0–242.0)	215.5 (190.5–258.5)	0.679
WBC	13.03 (7.97–16.38)	10.69 (6.74–15.50)	0.100
MCV (fL)	79.1 (40.15–80.4)	78.25 (74.65–80.4)	0.931
RDW (%)	15.3 (13.9–15.8)	14.7 (12.6–15.37)	0.147
PLT	306.0 (274.0–384.0)	373.0 (295.75–408.75)	0.339
NRBC	3.8 (2.0–5.4)	3.9 (2.12–5.72)	0.761
IG	3.6 (1.95–9.6)	3.85 (2.32–5.87)	1.000

Mann–Whitney U-test (median of variables) Mg: Magnesium, WBC: Wight Blood Cell, MCV: Mean Corpuscular Volume, RDW: Red cell Distribution Width, PLT: Platelet, NRBC: Nucleated Red Blood Cell, IG: Immature Granulocyte.

**Table 5 children-10-00528-t005:** The complete blood count results of children study and control group.

Variables	Total *n* = 122	Study Group *n* = 61 (50%)	Control Group *n* = 61 (50%)	*p*-Value
WBC	11.39 (7.44–15.86)	11.76 (7.44–16.19)	11.21 (7.36–14.56)	0.464
MCV (fL)	79.1 (73.1–80.4)	79.1 (72.2–80.4)	79.2 (751.5–805.5)	0.421
RDW (%)	13.4 (12.6–15.2)	14.9(13.0–15.7)	12.8 (12.3–13.7)	**<0.001**
PLT	315.0 (285.75–395.25)	315.0 (284.5–398.5)	315.0 (284.5–398.0)	0.900
NRBC	4.15 (1.77–12.67)	3.8 (2.15–5.55)	8.6 (1.45–1.49)	0.667
IG	5.2 (14.22–2.27)	3.8 (2.15–5.85)	11.5 (3.0–15.5)	**0.010**

WBC: Wight Blood Cell, MCV: Mean Corpuscular Volume, RDW: Red cell Distribution Width, PLT: Platelet, NRBC: Nucleated Red Blood Cell, IG: Immature Granulocyte.

## Data Availability

The data supporting the results analyzed and reported during the study can be accessed through the Kastamonu University Sisoft system (http://10.137.1.18:8180/sisoft/index.jsp?lang=TR&hasErr=true&type=noacc).

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
