# Peer review of "Comparison of Serum Selenium, Homocysteine, Zinc, and Vitamin D Levels in Febrile Children with and without Febrile Seizures: A Prospective Single-Center Study"

_children, 2023, doi:10.3390/children10030528_

Round 1
Reviewer 1 Report
1- It it better to use febrile seizure which is a well known phrase than Febrile convulsion (FC) among pediatric neurologist, and pediatricians.
2-Febrile convulsion (FC) is a convulsion accompanied by fever (> 38oC or >100.4 oF) due to any infection other than central nervous system (CNS) in children aged six months to five years without a history of febrile seizures or CNS anomaly (1,2)
May need to rephrase this statement.
Low levels of selenium, zinc, and vitamin D levels were found in patients with febrile seizures and normal serum selenium, zinc, and vitamin D levels may be protective to prevent the development of convulsions in children between the ages of 1-5 in the pediatric emergency department due to fever.
Author Response
lütfen yazın

Reviewer 2 Report
Dear authors,
I have now completed the review of the manuscript titled "Comparison of serum selenium, homocysteine, zinc, and vitamin D levels in febrile children with and without febrile convulsions: A prospective single-center study."
In the present study, the authors compared children with febrile convulsions to children without convulsions in blood selenium, zinc, homocysteine, and vitamin D levels.
The manuscript is interesting and, in general, fair written.
I have some suggestions to further improve the quality of the manuscript.
1. The introduction section seems to be relatively short. Please explain the results more or summarize other prior researches also.
2. I suggest authors clarify how other researchers can obtain the original data.
3. Authors used real world children data. Do they have disease history or family history also? Please describe.
4. In the ‘Statistical analysis’ section, the paragraph is quite long but there are no references for selecting your method. Please justify yourself about selecting those methods, or refer to some statistical standard and guidelines on methods for testing statistical differences between groups in medical research.
5. What is the future scope of the proposed research, authors have described the limitations in a good way, and I suggest that these can be the future scope of the work.
Author Response
Please write down

Reviewer 3 Report
Sample size calclation: The type of outcome should be specified (i.e. quantitative, continous).
Table 2: Freuquencies regarding children's sex should be presented to show that there is no relevant difference between the two groups (as mentiond in the text,
Statistical analysis: "All continuous variables were not showed a normal distribution": The structure of this sentence is not correct. Furthermore, medians should be presneted together with quartiles or with minimum and maximum.
Graphics 1 and 2: You should wirte in the legend that for each child more than one health condition issue could be specified.
Findings: For each health condition issue, absolute and relative frequencies should be give. The percentages should be given with the same number of decimal places. Inthe control group, 20% had a acute tonsillitis. How is this possible with n = 61? It may be interesting to compare the groups regarding each issue with simple Chi2 oder Fisher's exact tests.
Tables 4 and 5: What does "test" mean? Is this the rest statistic of the U test? I think this is not informative for the readers of the paper and may cause even cause confusion. For instance: The "test" for Vitamin D whoch is hoghly significant is rather high, the "test" for Selenium which is significant as well is rather low. Therfore I think, the "test" information is not important.
Tables 4 and 5: What do the numbers in parentheses represent?
All values should be given with 3 decimal places (i.e. 0.100 instead of 0.1 in table 5).
Statistical analysis: There are 4 variables with a statistically significant association. A mulitple logistic regression analysis with the binary outcome "febrile seizure" should be perfomed in order to find which of these varaibles is most important.
Concluision: This sentence should be worded more carefully.
Author Response
Please review the attachment

Round 2
Reviewer 3 Report
My suggestions for improvement have been considered and incorporated in the mansucript.